# Thymoquinone Alterations of the Apoptotic Gene Expressions and Cell Cycle Arrest in Genetically Distinct Triple-Negative Breast Cancer Cells

**DOI:** 10.3390/nu14102120

**Published:** 2022-05-19

**Authors:** Getinet M. Adinew, Samia S. Messeha, Equar Taka, Ramesh B. Badisa, Lovely M. Antonie, Karam F. A. Soliman

**Affiliations:** Division of Pharmaceutical Sciences, Institute of Public Health, College of Pharmacy and Pharmaceutical Sciences, Florida A&M University, Tallahassee, FL 32307, USA; getinet1.mequanint@famu.edu (G.M.A.); samia.messeha@famu.edu (S.S.M.); equar.taka@famu.edu (E.T.); ramesh.badisa@famu.edu (R.B.B.); lovely1.antoine@famu.edu (L.M.A.)

**Keywords:** breast cancer, triple-negative breast cancer, thymoquinone

## Abstract

Breast cancer (BC) is the most common cancer in women worldwide, and it is one of the leading causes of cancer death in women. triple-negative breast Cancer (TNBC), a subtype of BC, is typically associated with the highest pathogenic grade and incidence in premenopausal and young African American (AA) women. Chemotherapy, the most common treatment for TNBC today, can lead to acquired resistance and ineffective treatment. Therefore, novel therapeutic approaches are needed to combat medication resistance and ineffectiveness in TNBC patients. Thymoquinone (TQ) is shown to have a cytotoxic effect on human cancer cells in vitro. However, TQ’s mode of action and precise mechanism in TNBC disease in vitro have not been adequately investigated. Therefore, TQ’s effects on the genetically different MDA-MB-468 and MDA-MB-231 human breast cancer cell lines were assessed. The data obtained show that TQ displayed cytotoxic effects on MDA-MB-468 and MDA-MB-231 cells in a time- and concentration-dependent manner after 24 h, with IC_50_ values of 25.37 µM and 27.39 µM, respectively. Moreover, MDA-MB-231 and MDA-MB-468 cells in a scratched wound-healing assay displayed poor wound closure, inhibiting invasion and migration via cell cycle blocking after 24 h. TQ arrested the cell cycle phase in MDA-MB-231 and MDA-MB-468 cells. The three cell cycle stages in MDA-MB-468 cells were significantly affected at 15 and 20 µM for G0/G1 and S phases, as well as all TQ concentrations for G2/M phases. In MDA-MB-468 cells, there was a significant decrease in G0/G1 phases with a substantial increase in the S phase and G2/M phases. In contrast, MDA-MB-231 showed a significant effect only during the two cell cycle stages (S and G2/M), at concentrations of 15 and 20 µM for S phases and all TQ values for G2/M phases. The TQ effect on the apoptotic gene profiles indicated that TQ upregulated 15 apoptotic genes in MDA-MB-231 TNBC cells, including caspases, GADD45A, TP53, DFFA, DIABLO, BNIP3, TRAF2/3, and TNFRSF10A. In MDA-MB-468 cells, 16 apoptotic genes were upregulated, including TNFRSF10A, TNF, TNFRSF11B, FADD TNFRSF10B, CASP2, and TRAF2, all of which are important for the apoptotic pathway andsuppress the expression of one anti-apoptotic gene, BIRC5, in MDA-MB-231 cells. Compared to MDA-MB-231 cells, elevated levels of TNF and their receptor proteins may contribute to their increased sensitivity to TQ-induced apoptosis. It was concluded from this study that TQ targets the MDA-MB-231 and MDA-MB-468 cells differently. Additionally, due to the aggressive nature of TNBC and the lack of specific therapies in chemoresistant TNBC, our findings related to the identified apoptotic gene profile may point to TQ as a potential agent for TNBC therapy.

## 1. Introduction

Breast cancer (BC) is the most commonly diagnosed cancer in women worldwide [1], and it is one of the leading causes of cancer death in women [2]. According to the World Health Organization, BC became the most globally common cancer in 2021, accounting for 12% of all new cancer cases [3]. As of January 2022, more than 3.8 million women in the United States had a history of BC [4]. In the United States, 287,850 new invasive BCs are expected to be diagnosed in 2022, compared to 51,400 new non-invasive(in situ) BCs [5]. Triple-negative breast cancer (TNBC) cells lack the expression of the estrogen receptor (ER), progestin receptor (PR), and human epidermal growth factor receptor 2(HER2), which account for 15% to 20% of all newly diagnosed BC cases. TNBCs, which contain extremely unstable genomes, have been linked to several patient characteristics, including a higher predisposition for metastasis and more frequent diagnosis in younger women [6]. TNBC is a prominent topic in primary and clinical research for many reasons. TNBC is frequently associated with a higher pathogenic grade and incidence in premenopausal women and young African American women [7,8]. BC is the leading cause of death in Black women, assumed to be because of their higher vulnerability to develop TNBC than other ethnic groups, with one in every five Black women diagnosed with TNBC [4]. There is mounting evidence that there are significant racial differences in TNBC. According to previous studies, premenopausal African American (AA) women had a greater TNBC prevalence (34%) than premenopausal Caucasian American (CA) women (16%) [9]. Racial disparities in TNBC are closely associated with many biological and nonbiological factors [10]. For instance, compared with non-Hispanic white Americans, AA women diagnosed with TNBC had a higher tumor grade (65 vs. 43%), a more increased proliferation marker (KI67 > 10%, 88% vs. 54%), a higher inflammatory cytokine (IL-6, 4.5% vs. 0.88%), and a higher genetic mutation (TP53 46% vs. 27%; histone-lysine N-methyltransferase-MLL3 12% vs. 6%) [10]. TNBC has a worse prognosis when compared to non-TNBC, and most crucially, a lack of successful specific targeted therapy for TNBC [11].

Nowadays, chemotherapy is still the most common treatment for TNBC [12]. Doxorubicin and cisplatin are often used to treat TNBC [13]. However, these drugs develop acquired resistance and toxicity over time, leading to treatment limitations [14]. As a result, novel therapeutic approaches to tackle medication resistance and toxicity in TNBC patients are urgently required. 

Natural products, or their active constituents, are safe, affordable, and effective in treating various diseases, including cancer. Natural compounds are ubiquitously distributed throughout plants and marine organisms [15]. Black cumin (*Nigella sativa* L.) seed is one of the most promising natural products and has been used for over a thousand years to treat various human ailments [16]. The pharmacological properties of black cumin include anti-neoplastic, anti-inflammatory, anti-oxidant, anti-asthmatic, analgesic, anti-pyretic, anti-hypertensive, and anti-bacterial effects [17]. Meanwhile, the black seed is consumed in its natural form, and research into the primary components of the oil is underway to determine its mechanism of action. Out of all of these components, TQ has stood out as the most promising compound in preventing and treating different disorders, including cancer [18]. Figure 1 depicts the chemical structure of TQ.

TQ has been studied in in vitro and in vivo models for its antioxidant, anti-inflammatory, and anti-cancer properties [19]. Recently, myriads of research demonstrated the anticancer effect of TQ on several types of cancer, including breast, ovarian, larynx, colon, myeloblastic leukemia, osteosarcoma, and lung cancer [20,21]. Anti-proliferation, apoptosis induction, cell cycle arrest, and anti-metastasis/anti-angiogenesis are crucial mechanisms employed by TQ against cancer [22]. 

Finding innovative lead compounds that impede the proliferation of TNBC cells would be of therapeutic significance given the scarcity of effective treatments for TNBC. The TQ effect on TNBC, on the other hand, has not been adequately addressed. As a result, the current study investigated and compared TQ’s potential anticancer mechanisms on MDA-MB-231 and MDA-MB-468 cell lines. The cell growth, proliferation, invasion, migration, cell cycle progression, and apoptosis of two different TNBC cell lines were investigated in this study. In this study, we hypothesized that the response of the two genetically distinct cell lines to TQ would be exhibited via different apoptosis-related signaling pathways by influencing the expression of various genes controlling these events.

## 2. Materials and Methods

### 2.1. Materials and Reagents

TQ (purity ≥ 99%, cat # MKCC0600) and Alamar Blue^®^ (a resazurin fluorescence dye solution) were acquired from Sigma-Aldrich (St. Louis, MO, USA). Trypsin-EDTA solution, penicillin/streptomycin, and phosphate-buffered saline (PBS) were purchased from the American Type Culture Collection (ATCC; Manassas, VA, USA). An Annexin VFITC Apoptosis Detection Kit Plus (cat. no. 68FT-Ann VP-S) was purchased from RayBiotech (Norcross, GA, USA). Propidium Iodide Flow Cytometry Kit (cat. no. ab139418) was purchased from Abcam (Cambridge, MA, USA). A DNA-free™ kit (cat. no. AM1907) was purchased from Life Technologies, Inc. (Thermo Fisher Scientific, Inc., Waltham, MA, USA). An iScript™ cDNA Synthesis kit (cat. no. 170-8890), SsoAdvanced™ Universal SYBR^®^ Green Supermix, and the Human Apoptosis PCR array (Cat. no. 10034106) H96 were purchased from BioRad Laboratories (Hercules, CA, USA). Dulbecco’s modified Eagle’s medium (DMEM) and heat-inactivated fetal bovine serum (FBS) were purchased from VWR International (Radnor, PA, USA). Cell culture flasks were purchased from Santa Cruz Biotechnology, Inc. (Dallas, TX, USA). Cell culture plates were purchased from Thermo USA Scientific (Ocala, FL, USA). 

### 2.2. Cell Culture 

Two TNBC cell models, MDA-MB-231 (ATCC^®^ HTB-26™) and MDA-MB-468(ATCC^®^ HTB-132™), were purchased from ATCC and maintained following the company’s guidelines. Both cell lines were cultivated as monolayers in 75 mL tissue culture flasks at 37 °C in a humidified 5% CO_2_ incubator and subcultured when needed using trypsin-EDTA (0.25%). The complete growth DMEM contained 4 mM L-glutamine and was supplemented with 10% heat-inactivated FBS (*v*/*v*) and 1% penicillin/streptomycin salt solution (100 U/mL and 0.1 mg/mL, respectively). The experimental media were DMEM supplemented with 2.5% heat-inactivated FBS [23]. 

### 2.3. Cytotoxicity Assay Using Alamar Blue 

The cytotoxicity of TQ on TNBC cells was determined in MDA-MB-231 and MDA-MB-468 cells using Alamar Blue^®^ [24]. Cells were plated at a density of 5 × 10^5^ cells/100 µL/well in 96-well plates and incubated overnight at 37 °C. TQ was solubilized in DMSO, and both cell lines were treated for 24 h with the compound at concentration ranges from 0 to 50 μM. Wells treated in the same manner but without cells/compounds were used as blanks. The experiments were performed in triplicates. After a specified exposure period, 20 μL Alamar Blue^®^ in a concentration of 0.5 mg/mL was added to each well and incubated for 4 h at 37 °C. The Alamar Blue reagent contains an active ingredient, resazurin, which is reduced by metabolically active cells to a fluorescent form, resorufin. The reduced resazurin dye was measured at an excitation/emission wavelength of 530/590 nm using a Synergy HTX Multi-Mode microplate reader (BioTek Instruments, Inc., Winooski, VT, USA). The level of fluorescent signal correlates with cell viability. Concentrations of TQ that inhibited cell viability by <30% were selected for further experiments according to FDA-recommended guidelines for investigating diverse biological impacts [25].

### 2.4. Cell Morphological Assessment

The morphological changes in MDA-MB-231 and MDA-MB-468 TNBC cells were observed using a modified version of a previously described protocol [26]. On a 6-well plate, 5 × 10^5^ cells/mL were cultured and treated with or without TQ for 24 and 48 h at 5, 10, and 15 µM. After the incubation period, the medium was removed, and the cells were washed in PBS. The morphological changes in TNBC cells were seen at 40× magnification using a Cytation5 image reader phase-contrast microscope (BioTek, Instruments, Inc., Winooski, VT, USA).

### 2.5. Migration Assay

A linear scratch was made in a 12-well plate using a culture insert of 2 wells, separated by a wall of 500 µm (Cat# 80206, ibidi GmbH, Lochhamer Schlag 11, 82,166 Grafelfing, Germany). MDA-MB-231 and MDA-MB-468 TNBC cells were seeded at a density of 5 × 10^4^ cells/mL (70 µL of cell suspension) in each compartment and incubated overnight at 37 °C. On the next day, the culture insert was removed, and cell debris was washed away with a fresh medium. The compound concentration, which resulted in cell survival of more than 70% in cytotoxicity assays (i.e., 5, 10, and 15 µM of TQ), was utilized to examine the inhibitory effects of TQ against TNBC cell migration. Cells were then incubated for 0, 24 and 48 h with or without treatments. A vehicle control condition was adopted, which consisted of simple growth medium and DMSO. There were three copies of each replicate. Images from 5 different areas in each well were captured at 0, 24, and 48 h at a magnification of ×10. The distance between wound edges was measured using a Cytation5 cell Imaging reader (BioTek Instruments, Inc., Winooski, VT, USA). We calculated the area using ImageJ software (v1.50) [27]. Initial gap area (A1) and the final gap area after incubation were calculated in (A2) [28,29]:$$\left(1-\frac{A2}{A1}\right)*100$$

The difference between migration of vehicle control cells and treated cells was used to quantify TQ’s prevention of cancer cell migration, i.e., migration inhibition: $$\left(\frac{A2\left(\mathrm{treatment}\right)}{A1}\right)-\left(\frac{A2\left(\mathrm{control}\right)}{A1}\right)$$

### 2.6. Cell Invasion Assays

The TNBC cell invasion assay was carried out in a Corning^®^ Matrigel^®^ Invasion Chamber 6-well plate (8.0 μm pore size; Discovery Labware, Inc., Bedford, MA, USA) using previously described protocols with certain modifications [30]. The Matrigel in the transwell chamber was rehydrated with 600 uL prewarmed complete growth media for 2 h in the incubator. After 2 h, the media were removed, and 200 μL of medium with 2.5% fetal bovine serum (FBS) containing 1 × 10^5^ cells was added to the upper chamber and incubated for 2 h to settle the cells in the membrane, then treated with TQ at different concentrations. The medium with 20% FBS (1 mL) in the lower chamber was used as a chemoattractant. Then, the cells were incubated at 37 °C for 24 h. The cells remaining on the upper chamber surface were removed using a cotton swab. Cells that moved into the lower surface of the filters were fixed with 70% ethanol for 10 min, followed by 10 min staining with 0.2% crystal violet. The invasive cells were observed and calculated under five fields at random using a Cytation5 cell imaging reader (BioTek Instruments, Inc., Winooski, VT, USA). This experiment was conducted in triplicate.

### 2.7. Cell Proliferation Assay

The effect of TQ on cell proliferation in MDA-MB-231 and MDA-MB-468 TNBC cells was determined using the Alamar Blue^®^ test [31]. Briefly, cells were seeded in 96-well plates (1 × 10^4^ cells/100 µL/well) and incubated at 37 °C overnight. In a final volume of 200 µL/well, both MDA-MB-231 and MDA-MB-468 cell lines were treated with TQ at the same concentrations ranging from the cytotoxicity assay (0–50 µM) for 48, 72, and 96 h. As a blank, equivalent wells without cells were employed. At the end of each exposure period, 20 µL of Alamar Blue^®^ was added to each well, the plates were incubated for 4 h at 37 °C, and the plates were read using a Synergy HTX Multi-Mode microplate reader (BioTek Instruments, Inc., Winooski, VT, USA).

### 2.8. Colony Formation Assay

A clonogenic assay was performed to measure the long-term effect of TQ on MDA-MB-231 and MDA-MB-468 TNBC cells using the previously published procedure [32]. Briefly, both cell lines were seeded at a density of 1 × 10^4^ cells/100 µL/well and treated in 96-well plates in the same way as the previous proliferation investigation. After the exposure period (3 or 6 h), the media containing DMSO/TQ were aspirated, and all wells were rinsed twice with PBS. The cells were then allowed to grow for seven days, after which the colonies generated in both treated and untreated cells were assessed using the Alamar Blue^®^ reduction assay previously described in the cell viability assay. Each assay was performed in triplicate.

### 2.9. Cell Cycle Analysis

Following previously established techniques, the effect of TQ on DNA content and cell cycle distribution was assessed for both MDA-MB-231 and MDA-MB-468 cell lines [33]. Both cells were seeded overnight at 1.5 × 10^6^ cells/4 mL/T-25 cell culture flasks and treated with TQ at 0, 10, 15, and 20 µM in a final volume of 6 mL/flask of experimental medium. Both floating and attached cells were collected after 24 h treatment, pelleted, washed in PBS, and fixed in cold 70% ethanol. The cells were again centrifuged for 5 min at 3000 rpm in a low-temperature centrifuge, rinsed in PBS, and gently resuspended in 200 µL PBS with 1× propidium iodide (PI = 0.02 mg/mL) +RNase (0.1 mg/mL) staining solution, and incubated for 30 min at 37 °C in the dark. Using a Sony SH800 cell sorter (Sony biotechnology, San Jose, CA, USA), the distribution of DNA in all cell cycle phases was measured in replicates within 2 h. Each sample contained a total of 20,000 individual events. Data were collected using the SH800 software, and the percentage of cells in each phase was calculated with the ModFit LT 5.0 software (Verity software house Inc., Topsham, ME, USA). GraphPad Prism was used to examine the statistically significant effect of TQ on cell cycle distribution at different stages of the cell cycle-G0/G1 phase, S phase, and G2/M phase [34].

### 2.10. Apoptosis Assay

The apoptotic effect of TQ was assessed in MDA-MB-231 and MDA-MB-468 cells using the previously published procedure [32]. MDA-MB-231 and MDA-MB-468 cells were plated in 6-well plates (5 × 10^5^ cells/2 mL/well) and incubated overnight at 37 °C. Cells were similarly treated with TQ at concentrations ranging from 0 to 30 µM in a final volume of 4 mL/well of experimental media to investigate the apoptotic effect of the compound. Control cells were exposed to only experimental media with DMSO. The treated and control cells from each well were collected, pelleted, and washed in PBS after a 24 h exposure period. The cell pellets were then suspended in 500 µL of 1 × Annexin V binding buffer before being labeled with 5 µL each of Annexin V-FITC and PI. The apoptotic effect was measured in replicates for 5–10 min using a Sony SH800 cell sorter (Sony Biotechnology, San Jose, CA, USA). Each sample had a total of 10,000 individual events. SH800 software was used to collect the data, and the ModFit LT 5.0 software (Verity software house Inc., Topsham, ME, USA) was used to calculate the percentage of cells in each phase. TQ’s statistically significant effect on alive cells, apoptotic cells, and necrotic cells was investigated using GraphPad prism.

### 2.11. Gene Expression Assay

Profiling of the various apoptotic gene expression was established in MDA-MB-231 and MDA-MB-468 TNBC cells using previously described methodologies [32]. In brief, each cell line (at a density of 6 × 10^6^ cells/10 mL in T-75 flasks) was incubated overnight at 37 °C, then treated with TQ for 24 h in a final volume of 20 mL/T-75 flasks. At concentrations near the IC_50_ values, TQ was chosen to treat the cells. After 24 h treatment, cells were mechanically collected from each flask, pelleted, and washed twice with PBS. According to the manufacturer’s instructions, total RNA was isolated from each cell pellet using 1 mL of TRIzol reagent. For phase separation, 0.2 mL of chloroform was added to each sample, vortexed, incubated at room temperature for 2–3 min, and centrifuged for 15 min at 10,000× *g* and 2–8 °C. The aqueous phase was collected in fresh centrifuge tubes and mixed with 0.5 mL of isopropyl alcohol to pellet the RNA. The RNA pellets were then washed with 70% ethanol, reconstituted in nuclease-free water (30 μL), and placed in an −80 °C freezer for later use. RNA quantity and purity were measured in each sample using a NanoDrop spectrophotometer (NanoDrop Technologies; Thermo Fisher Scientific, Inc., Waltham, MA, USA). Finally, the cDNA for each sample was synthesized using the iScript^TM^ cDNA Synthesis kit and was stored in a −80 °C freezer. Each well of the 96-well human apoptosis array was loaded with 10 μL each of the reconstituted cDNA (2.3 ng) and SsoAdvanced™ Universal SYBR^®^ Green Supermix, and the plate was placed for 5 min in a shaker and centrifuged at 1000× *g* for 1 min. The fluorescent quantitative PCR run was established using a Bio-Rad CFX96 Real-Time System (Bio-Rad Laboratories) with 39 thermo-cycling of denaturation as follows: 30 s activation at 95 °C, 10 s denaturation at 95 °C; 20 s annealing at 60 °C; and 31 s extension at 65 °C [35]. Three independent studies for each cell line verified the RT-PCR results. The CFX 3.1 Manager software (Bio-Rad Laboratories) was used to assess gene expression, and the results were confirmed using the Student’s *t*-test.

### 2.12. Statistical Analysis 

This study analyzed data from GraphPad Prism 9.3.1 software (GraphPad Software, Inc., San Diego, CA, USA). All data points present the average of at least three independent experiments are expressed as the mean ± S.E.M. The IC_50s_ values were calculated by the nonlinear regression model of log (inhibitor) vs. normalized response variable slope using the software, with the R2 best fit and the lowest 95% confidence interval. The significance of the difference was determined using a one-way or two-way analysis of variance (ANOVA) as indicated in the legends, followed by Bonferroni’s multiple comparison test. Unpaired Student’s *t*-test was used for comparing two data sets. Generally, a difference was considered significant at * *p* < 0.05 (as indicated in the figures and legends).

## 3. Results

### 3.1. TQ Induces Cytotoxicity in Triple-Negative Breast Cancer Cells in a Concentration and Time-Dependent Manner 

The TNBC cell lines, MDA-MB-231 and MDAMB-468, were treated at different concentrations of TQ, 0–50μM. The cytotoxicity effect was evaluated by an Alamar blue assay. The effect of TQ on MDA-MB-231 and MDA-MB-468 cell viability is shown in Figure 2A,B. In parallel with MDA-MB-231 cells, a reduction in cell viability was found in MDA-MB-468 cells, as indicated in Figure 2A,B (IC_50_ = 27.39 ± 0.13 μM (4.5 µg/mL) for MDA-MB-231 cells and 25.37 ± 0.36 μM (4.2 µg/mL) for MDA-MB-468 cells) at 24 h exposure to TQ. These changes indicate a higher sensitivity of this cell line to TQ. Our data, after 12 and 24 h treatment, showed that TQ reduced the TNBC cell line viability in a time- and concentration-dependent manner. TQ concentrations at 5 and 10 µM had no significant effect on the viability of MDA-MB-231 and MDA-MB-468 cells after a 24 h exposure period. The viability of the MDA-MB-231 cell was reduced by 4%, 19%, and 68% after 24 h of incubation at concentrations of 10, 20, and 30 µM of TQ, respectively (Figure 2A). Cell viability in MDA-MB-468 cells was reduced by 3%, 38%, and 59% after 24 h of incubation at 10, 20, and 30 µM of TQ, respectively (Figure 2B). Since DMSO was used as a solvent, its effect on these two cells was evaluated, and no significant adverse effects were detected. Anti-proliferative tests were used to assess the long-term effect of TQ on MDA-MB-231 and MDA-MB-468 TNBC cells, as evidenced by the growth-inhibitory potency observed with longer exposure times. Overall, the results showed that the proliferation rate was reduced in a concentration- and time-dependent manner. TQ significantly suppressed cell growth in both cell lines after 48, 72, and 96 h of treatment compared to each exposure period (*p* < 0.0001). In the MDA-MB-231 cell (Figure 2C), at a concentration of 10 µM, the proliferative rate was reduced by 18, 29, and 35% at 48, 72, and 96 h exposure periods, respectively. For its counterpart, MDA-MB-468 cell (Figure 2D), the proliferation rate reduced by 73, 76, and 85%, respectively, at the same concentration. Overall, similar to the results of the viability investigation, the MDA-MB-468 cell line response to the anti-proliferative impact of TQ was higher than MDA-MB-231 cells. These disparities in TQ’s responses could point to distinct molecular pathways.

### 3.2. TQ Alters the Morphology of TNBC Cells in a Concentration- and Time-Dependent Manner

MDA-MB-231 and 468 TNBC cell lines 5 × 10^5^ cells were transferred to 6-well plates and cultured in an incubator for 24h. Both cancer cell lines were treated with TQ at 5, 10, and 15 µM at 24 and 48 h. After TQ acted, the morphology of TNBC cell lines changed relative to the control group. The microscopic observation of TQ-treated TNBC cells showed a reduced cell density with increasing TQ concentrations. The cells became smaller, irregular aggregates with a rounded, granulated appearance, breaking into fragments. The control cells did not show any morphological changes, which have a uniform layer of cells with many tight connections between neighboring cells (Figure 3). 

### 3.3. TQ Inhibits the Clonogenicity of Triple-Negative Breast Cancer Cells

We next examined the potential of TQ to impact clonogenic cell survival. Exposure of the cells at 3 and 6 h to 0–50 µM resulted in a significant inhibition of clonogenic cell survival. The effect of TQ on TNBC cell colony formation was demonstrated by a considerable reduction in the number of colonies in a time- and concentration-dependent manner. Indeed, treating MDA-MB-231 cells for 3 h with a gradient concentration of TQ caused a significant 21% decrease (*p* < 0.01) in colony formation at 10 µM, then dramatically reduced to 82% at 15 µM (*p* < 0.0001). Interestingly, increasing the exposure period to 6 h showed a minor difference, at 10 µM, compared with the 3 h treatment period, which was reduced by 41% (Figure 4A). In contrast, particularly at 10 µM, a highly significant (*p* < 0.0001) anti-clonogenic effect (26% inhibition) was found in MDA-MB-468 cells after three hours of exposure; meanwhile, there was a gradual decrease with increasing TQ concentration, exhibiting a significant 90% inhibition of colony formation at the highest tested concentration, 50 µM. In the same line, a significant difference was measured between a 3 h vs. 6 h exposure period, as indicated by a more than 40% and 89% reduction in colony formation at 5 µM and 10 µM, respectively (Figure 4B). The results were consistent with the findings in the cell viability as well as the proliferation assays and further confirmed the analogous responses of TNBC cells to TQ treatment.

### 3.4. TQ Decreases Cell Migration in a Time- and Concentration-Dependent Pattern

We further evaluated the effect of TQ on TNBC cell migration using culture inserts of 500 µm diameter. For this assay, MDA-MB-231 and MDA-MB-468 cells were identically treated with TQ at 5, 10, and 15 µM along with their respective untreated controls. Images were captured at 0, 24 and 48 h to measure the changes under different treatment conditions. The results of both cell lines displayed a highly significant repression of cell migration in a time- and concentration-dependent manner when cells were exposed to low concentrations of TQ. 

Our result revealed that, in MDA-MB-231 cells (Figure 5B,C), TQ lowers migration by 15, 23, and 26% at 5, 10, and 15 µM, respectively, after 24 h (*p* < 0.05–0.01). After 48 h, migrating cells were reduced by 25, 40, and 47%, respectively, at the same concentration (*p* < 0.001). After 24 h, a minor reduction in cell migration was exhibited by TQ-treated MDA-B-468 cells, reaching only 5% at the highest tested concentration (15 µM) (Figure 5E,F), compared to control; at the same time, there was no significant difference in migration at 5 and 10 µM. A significant (*p* < 0.001) and gradual repression of cell migration was measured after 48 h exposure period to the compound. At 5, 10, and 15 µM, the migrating cells were inhibited by 36, 42, and 47 percent after 48 h, respectively, compared to the control group. According to our findings, we propose TQ is a successful compound for reducing the cell migration of TNBC cells and suggesting that TQ could be an essential molecule in the prevention and treatment of TNBC metastasis. The migration study in both cell lines, particularly MDA-MB-468 cells, showed that the gap at 15 µM concentration of TQ after 48 h increased the gap of the wound compared to 0 h, which could be an indication of reversing the invaded cells.

### 3.5. TQ Decreases Cell Invasion in a Concentration-Dependent Manner 

The migration of TNBC cells was originally tested in a wound-healing assay. TNBC-MDA-MB-231 and MDA-MB-468 cell migration was decreased by treatment with 5, 10, and 15 µM (Figure 5). TQ’s anti-invasive action was further investigated by utilizing transwell chamber invasion experiments. TQ reduced the number of cells on the outside of the transwell membrane in a concentration-dependent manner after 24 h of treatment. In MDA-MB-231, TQ concentrations of 5,10, and 15 µM reduced the number of invasive cells in the outer membrane by 14, 17, and 19 percent, respectively, when compared to control (Figure 6A–E). At the same concentrations, the invasive cells decreased by 3, 17, and 54 percent in MDA-MB-468 cells, respectively (Figure 6F–J). These data indicated that invasion of TNBCs was inhibited by TQ treatment. 

### 3.6. Thymoquinone Induces Cell Cycle Arrest in Triple-Negative Breast Cancer Cells

Sony SH800 cell sorter analysis utilizing PI labeling was used to examine the cell cycle distribution in MDA-MB-231 and MDA-MB-468 cell lines after 24 h of exposure to 0, 10, 15, and 20 µM of TQ into determine the mechanism underlying the cytotoxic and anti-proliferative effects of TQ. The findings, as shown in Figure 7, demonstrate that, at these exposure levels, TQ therapy had a significant effect on the cell cycle distribution in both cell lines. MDA-MB-468 cells were more vulnerable to TQ than MDA-MB-231 cells, even though both models behaved similarly at the S-phase, with the S-phase increasing with increasing concentration. The three cell cycle stages in MDA-MB-468 cells were significantly affected (*p* < 0.05–*p* < 0.01) at 15 and 20 µM for G0/G1 and S phases, as well as all TQ concentrations for G2/M phases. In MDA-MB-468 cells, there was a significant decrease in G0/G1 phases with a substantial increase in the S phase and G2/M phases (Figure 7G,J). According to the data, the proportions of S-phase cells, and thus S-phase arrest, were more prominent in the treated cell group than in the control group. In contrast, MDA-MB-231 showed a significant effect only during the two cell cycle stages (S and G2/M), at concentrations of 15 and 20 µM for S phases and all TQ values for G2/M phases. At 20 µM, the percentages of S phase arrest cells were slightly higher in MDA-MB-468 cells compared with MDA-MB-231 cells, as shown in Table 1 and Figure 7F–J (21.95 ± 2.7 vs. 30.67 ± 2.7 in MDA-MB-231 and 22.8 ± 1.4 vs. 34.7 ± 2.1 in MDA-MB-468, when compared with the DMSO-treated control). This behavior was exhibited, along with a significant decrease in G2/M-phase cells by 9% (12.1 ± 2.4 vs. 3.3 ± 1.3, *p* < 0.01) in MDA-MB-231 cells and a substantial increase in G2/M phase cells in MDA-MB-468 cells by 9% (0.9 ± 0.6 vs.10.9 ± 2.4; Table 2). These events were observed in MDA-MB-231 cells at the highest concentration, accompanied by a rise in the number of dead cells (Sub G1), as shown in the G0/G1 peaks left in Figure 7D. In MDA-MB-468 cells, at the medium and highest concentrations (15 and 20 µM) as shown in Figure 7C,D,H,I, a concentration-dependent increase in such a peak was observed, resulting in a significant increase in the number of apoptotic cells (Sub G1). When paired with data generated from the proliferation study, these findings pointed to a distinct TQ mechanism in MDA-MB-468 that was more effective than the mechanisms that influenced the MDA-MB-231 cell line.

The mean distribution of G0/G1, S-phase, and G2/M phases in treated and control TNBC cells after 24 h were presented in Table 1 and Table 2, for MDA-MB-231 and MDA-MB-468 cells, respectively. 

The mean distribution of cell cycle phases in treated and control TNBC cells after 24 h exposure to TQ at various concentrations for MDA-MB-231 and MDA-MB-468 TNBC cells is shown in Table 1 and Table 2, respectively. TQ has a concentration-dependent effect on the cell cycle phase in both cell lines. The *p*-value for the difference between the control and treated cells at different cell cycle phases was determined using one-way ANOVA followed by Bonferroni’s multiple comparisons test. At * *p* < 0.05, ** *p* < 0.01, the difference was declared significant.

### 3.7. Thymoquinone Induces Apoptosis in Triple-Negative Breast Cancer Cells

TQ-treated cells were more or less damaged by TQ, as evidenced by morphological findings. To better understand the proportions of apoptosis and necrosis in TQ-treated TNBC cell lines, a Sony SH800 cell sorter apoptosis study was performed using Annexin V-FITC/PI double labeling. In both cell lines, a significant (*p* < 0.05–*p* < 0.0001) concentration-dependent apoptotic impact was detected after a 24 h exposure period to TQ at concentrations ranging from 0–30 µM (Figure 8). Similar to the viability, proliferation, clonogenic, migration, and invasion experiments, the apoptotic impact in MDA-MB-231 cells was slower when compared to the TQ-sensitive MDA-MB-468 cell line. When MDA-MB-468 cells were treated with 15 µM TQ, 56 percent of the cells entered the apoptotic and necrotic phases, while 53 percent of MDA-MB-231 cells entered the apoptotic and necrotic phases at the same concentration. TQ treatment at various concentrations (5, 10, 15, 20, 25, and 30 µM) resulted in increased populations of apoptotic and necrotic cells in both cell lines compared to control. There were concentration-dependent increases in the number of apoptotic cells and a decrease in viable cells. In MDA-MB-231 cells (Figure 8A–G), TQ at the lowest (5 µM) and highest tested concentrations (30 µM) induced an early apoptotic effect of 2.48% and 28.79%, respectively. The late apoptotic cell population was 1.88% and 23.14%, whereas, in MDA-MB-468 cells (Figure 8I–O) at the same concentrations, the early apoptotic cell populations were 7.40% and 8.94%, and the late apoptotic cell populations were 0.8% and 36.87, respectively. In these cells, TQ had a more substantial anti-proliferative effect. In conclusion, TQ was shown to cause apoptosis and necrosis in TNBC cell lines, suggesting that TQ appears to be a promising candidate for future research.

### 3.8. Apoptotic Gene Expression Alteration in TQ-Treated Triple-Negative Breast Cancer Cells

To investigate the anticancer mechanism of TQ, TQ-treated TNBC cells were subjected to quantitative real-time PCR to determine the transcriptome level of apoptosis-related genes. Each cell line was given different concentrations based on the calculated IC_50_ values (27 µM in MDA-MB-231 cells and 25 µM in MDA-MB-468 cells, respectively, as shown in Figure 2A,B. TQ’s impact on numerous apoptosis-linked genes was inferred by profiling normalized mRNA expression for the cells under study; however, we only showed the significantly altered mRNAs (Figure 9A,B). The red dots represent upregulated genes; the green dots only in MDA-MB-231 cells reflect downregulated genes, and the black dots in both cell lines indicate that gene expression is unchanged. Figure 9C,D show a bar graph comparing the expression of various apoptosis regulating genes on both TNBC cells when the treated group was compared to the corresponding control group. TQ induced the expression of many essential genes that control apoptosis in TNBC cell lines. TQ significantly increased the expression of Caspases 3, 4, and 9 in MDA-MB-231 cells (*p* < 0.05–*p* < 0.01), but only Caspase 2 (*p* < 0.05) was upregulated in MDA-MB-468 cells. TRAF2, BAG1, ACTB, and TNFRSF10A were found to be overexpressed in both cell lines, with TRAF3 being overexpressed in the MDA-MB-231 cell line **(**Figure 9C,D). Furthermore, when the mRNA profiles of both cell lines were compared, TQ treatment was found to have an effect on a subset of specific genes in only one cell line. After 27 µM of TQ treatment, BNIP3, BCL10, TP53, DIABLO, MCL1, DFFA, and GADD45A were upregulated, specifically in MDA-MB-231 cells, and BIRC5 gene expression was repressed (Figure 9C). TQ increased the expression of FADD, TNF, TNFRSF (10A, 11B and 21), BAX, DAPK1, APAF1, AKT1, BIK, and CASP2 genes only in MDA-MB-468 cells after exposure to 25 µM (Figure 9D).

Surprisingly, in terms of fold-change, MDA-MB-468 cells outperformed MDA-MB-231 cells (Table 3B). As a result, the findings may help to explain why TQ has a higher anti-cancer activity against MDA-MB-468 cells in our cytotoxicity, proliferation, colony formation, migration, and cell cycle assay results. Among the upregulated genes, 15 genes showed a significant increase (*p* < 0.05–*p* < 0.001) in their mRNA level (between 2.98 and 27.67-fold) after MDA-MB-468 was exposed to 25 µM of TQ. TNFRSF10A had the highest fold increase (27.67-fold) followed by TNF (+26.07), and DAPK1 (2.98-fold) had the lowest fold increase among these upregulated genes (Table 3A). When MDA-MB-231 cells were exposed to 27 µM TQ, the mRNAs of several apoptotic genes were altered, and 15 genes were significantly impacted with 1.8–4.5-fold upregulation: GADD45A had the highest fold increase (4.5-fold), and CASP3 had the lowest fold increase (1.8-fold). In MDA-MB-231 cells, BIRC5 was the only gene that was downregulated (3.58-fold) (Table 3A).

In MDA-MB-231 (Table 3A), actin, beta (ACTB), BCL2-associated athanogene 1 (BAG1), B-cell CLL/lymphoma 10 (BCL10), BCL2/adenovirus E1B 19 kDa interacting protein 2 (BNIP3), DNA fragmentation factor, 45 kDa, alpha polypeptide (DFFA), diablo, IAP-binding mitochondrial protein (DIABLO), growth arrest and DNA-damage-inducible, alpha (GADD45A), Caspase (CASP3, CASP4, CASP9), myeloid cell leukemia sequence 1 (BCL2-related) (MCL1), tumor necrosis factor receptor superfamily, member 10A (TNFRSF10A), tumor protein p53 (TP53), TNF receptor-associated factor 2 (TRAF2), and TNF receptor-associated factor 3 (TRAF3) were upregulated, while baculoviral IAP repeat-containing 5 (BIRC5) was downregulated. In MDA-MB-468 cells (Table 3B), actin, beta (ACTB), v-akt murine thymoma viral oncogene homolog 1 (AKT1), apoptotic peptidase activating factor 1 (APAF1), BCL2-associated agonist of cell death (BAD), BCL2-associated X protein (BAX), BCL2-associated athanogene 1 (BAG1), BCL2-interacting killer (apoptosis-inducing) (BIK), death-associated protein kinase 1 (DAPK1), Caspase 2 (CASP2), Fas (TNFRSF6) associated via death domain (FADD), tumor necrosis factor (TNF), tumor necrosis factor receptor superfamily, member 10A (TNFRSF10A), tumor necrosis factor receptor superfamily, member 10b (TNFRSF10B), tumor necrosis factor receptor superfamily, member 11b (TNFRSF11B), tumor necrosis factor receptor superfamily, member 21 (TNFRSF21), and TNF receptor-associated factor 2 (TRAF2) were upregulated.

The significantly upregulated genes in both cell types were ACTB, BAG1, TNFRSF10A, and TRAF2 mRNA; yet the fold increase in MDA-MB-468 cells was higher than that in MDA-MB-231 cells, with 6.36 vs. 2.1 for ACTB, 11.38 vs. 2.7 for BAG1, 27.67 vs. 2.3 for TNFRSF10A, and 11.42 vs. 2.3 for TRAF2, respectively (Table 3). Our findings call attention to the distinct response mechanisms displayed by two genetically distinct TNBC cells in response to TQ.

## 4. Discussion

BC is a disease with a wide range of biological features, morphological patterns, and clinical manifestations [36]. TNBC is the most aggressive subtype, with frequent distant metastases, early recurrence, and a poor overall survival rate [37]. TNBC, which accounts for 15–25% of all BC types, is most common in young women under 40 [1]. Both endocrine treatment and HER-2 targeted therapy are ineffective [38]. Drug resistance significantly impacts the efficacy of chemotherapy, endocrine therapy, and targeted molecular therapy, leading to recurrence, metastasis, and potentially the death of TNBC patients [39]. 

TQ, an essential metabolite isolated from *N. sativa*, with a diversified medicinal effect, including anticancer effects, but these anticancer effects of TNBC are not extensively researched, with only a few studies to date. The results of our study show that TQ has anticancer properties in two different TNBC cell lines. TQ was discovered to have cytotoxic effects and to inhibit cell proliferation in both MDA-MB-231 and MDA-MB-468 cell lines. Cytotoxicity studies (Figure 2), proliferation assays (Figure 2), clonogenic assays (Figure 4), migration assays (Figure 5), invasion assays (Figure 6), and cell cycle distribution analysis (Figure 7) revealed that these two cell lines responded differently. The viability of MDA-MB 231 and MDA-MB-468 cells decreased after treatment with 0–50 µM of TQ for 12, 24, and 48 h compared to control. The effect was more pronounced when the TQ concentration was increased, with a significant decrease in the viability of both cells. The results show that MDA-MB-468 cells are more sensitive to TQ than MDA-MB-231 cells. TQ was found to prevent the growth and proliferation of a variety of cancers in in vitro research, including colorectal [40], prostate [41], leukemia [40], pancreatic, and lung cancer [42]. In this regard, the findings imply that TQ may be beneficial in the prevention and treatment of TNBC, a kind of BC that is highly proliferative.

A microscopic observation of TQ-treated TNBC cells showed a reduced cell density. Due to their cytoplasmic contraction, the cells become smaller, irregular aggregates with a rounded, coarse appearance and break into fragments. The treated cells started to shrink, and cell shrinkage progressively increased in a concentration-dependent manner. This shrinkage may be due to the growth inhibitory effect of TQ. Many apoptotic cells appeared, the cell space increased, and intracellular contacts disappeared [43]. Thus, the compound induced the apoptosis of MDA-MB-231 and MDA-MB-468 cells and inhibited proliferation. Decreased adherence in TQ-treated TNBC cells can lead to disorganized cell spreading, fewer cell–cell contacts, and the disheveled appearance of the cell population. The control group has a uniform cell layer with tight connections between neighboring cells. Treatment of cancer cells with natural compounds can alter their morphology, leading to ultra-structural perturbations. Changes in cytoskeleton integrity, destruction of the monolayer, membrane composition, endoplasmic reticulum, mitochondria, and the nucleus lead to the formation of morphological alterations, preceding cell death [44].

The wound-healing assay is low-cost and straightforward to perform, and it is often used to analyze cell mobility. Both cell migration and cell proliferation are crucial for wound closure [45]. Cell migration and invasion processes are critical for cancer cells to metastatically spread [46]. In TNBC patients, cancer cell migration is a significant cause of death [47]. TQ’s anti-metastatic capabilities have been under-studied despite extensive study on its usefulness in cancer treatment. Compounds with the ability to alter oncogenic signaling pathways have shown considerable progress in eradicating cancer cells and shrinking tumors [48,49]. However, subgroups of cancer cells resist therapy due to innate heterogeneity and treatment-induced drug adaptation [50]. In addition to local recurrence, these cells can leave a primary tumor and travel through the stroma to gain access to the bloodstream, where they can spread to other organs, resulting in an incurable disease. When cancer cells invade neighboring tissues, they reject signals from neighboring cells and continue to develop [51]. As a result, therapies that restrict or diminish cancer cell migration and invasion may improve cancer therapy by dramatically inhibiting or reducing metastasis. Targeting invasion and migration are especially essential for malignancies that currently lack specific treatments, such as TNBC [52]. Several studies have reported the role of TQ in cancer cell migration [53,54,55,56]. The authors noted that TQ inhibits the migration and invasion of prostate cancer [57] and BC [58] in a concentration-dependent manner. Based on our viability study, TQ was not harmful to TNBC cells at concentrations of 5 and 10 µM. Importantly, TQ had a statistically significant inhibitory effect on cell migration at these concentrations. This rule out the possibility that the anti-migratory effect was due to something other than the TQ concentration’s direct killing effect. Furthermore, the results of this study demonstrated that apoptosis, or programmed cell death, is not the only key mechanism preventing TNBC cells from migrating. We revealed that TQ has anti-migratory and anti-invasive properties on TNBC cells. When comparing the 0 h and 48 h treatment periods, the treated cells began to shrink, and cell shrinkage progressively increased in a concentration-dependent manner, particularly in MDA-MB-468 cells at a 15 µM concentration of TQ after 48 h, as demonstrated by the increasing size of the wound gap. This edge shrinkage could be due to TQ’s antimigration effect, which could be an indication that the invaded cells are being reversed. As a result, our findings suggest that TQ affects the migratory ability of TNBC cells.

Clonogenic assays assess a cell’s ability to maintain its integrity for an extended length of time [59]. Clonogenic assays are crucial because they display phenotypic changes that take time and sometimes many cell divisions to function. Because stem cells are long-lived cells with the ability to increase indefinitely, clonogenic assays are also employed to assess the stemness of specific cell types [60]. The clonogenic assay is especially important in TNBC research since cancer stem cells are frequently linked to chemoresistance, subsequent tumor development, and cancer recurrence [61]. As a result, using a clonogenic assay to investigate TNBC stemness is a valuable and commonly used tool for predicting the efficacy of a particular therapy. Our studies demonstrated that TQ significantly suppressed the colony development of TNBC over an extended length of time, indicating that it could be a viable therapy option for TNBC, which is challenging to treat with current conventional chemotherapeutic medicines.

Uncontrolled cell growth and proliferation are the hallmarks of cancer. Many variables influence this behavior, including the activation of oncogenes and the silencing of tumor suppressor genes. Overexpression of cyclin-dependent kinase (CDK) complexes and the inactivation of cyclin-dependent kinase inhibitors are two of the most common causes of cancer genesis (CKIs) [62]. Cell apoptosis occurs when particular checkpoints in the cell cycle are disrupted [63]. TQ reduces cell growth in several ways by regulating key cell cycle checkpoints and triggering cell cycle arrest at the G0/G1, G1/G2, or G2/M stages. Sony SH800 cell sorter analysis of TQ-treated TNBC cells revealed that TQ induces cell cycle arrest during G0/G1, S, and G2/M phases. Consistent with our findings, TQ causes cell cycle arrest in various cancer cells, including colorectal [64], prostate [65], liver [66], lung [67] and TNBC [68]. Furthermore, TQ-mediated S phase cell cycle arrest was linked to a significant reduction in the G2/M population, representing apoptosis-prone cells [42]. Taken together, according to our findings, TQ significantly induces cell cycle arrest at various stages of the cell cycle phases, which could be used as an alternative therapeutic option to stop the uncontrolled proliferation features of TNBC.

On the other hand, cell apoptosis is an essential regulator of cell proliferation. Regardless of the source or type, cancer is characterized by uncontrolled proliferation, angiogenesis, and apoptosis resistance. Loss of apoptotic regulation allows cancer cells to survive longer and acquire mutations, which can increase tumor invasiveness, cause angiogenesis, deregulate cell proliferation, and interfere with differentiation [69,70]. One technique for treating cancer is to control or stop the uncontrolled multiplication of cancer cells. So, the most effective non-surgical treatment is apoptosis induction. As a result, studying apoptosis inducers is viewed as a critical method for discovering anti-cancer drugs [71]. Our studies identified cell apoptosis upon exposing TNBC to TQ at 27 µM in MDA-MB-231 cells and 25 µM in MDA-MB-468 cells for 24 h. Our findings were consistent with previous studies in TP3 mutant TNBC cells, which demonstrated that TQ induces apoptosis [68]. TQ may effectively cause apoptosis in MDA-MB-231 and MDA-MB-468 cells in a concentration-dependent manner, showing that TQ could be an effective apoptosis inducer.

Multiple molecular pathways control apoptosis and coordinate cell proliferation and death to maintain homeostasis in normal cells. However, activating the genes implicated in these signaling pathways increases cell proliferation and carcinogenesis in cancer cells. Understanding the apoptosis processes and targeting the expression of these apoptotic genes is, therefore, critical for developing tailored cancer therapy [72]. In the current study, changes in gene expression patterns indicated TQ’s ability to trigger apoptosis via intrinsic and extrinsic mechanisms. TQ was found to trigger intrinsic and extrinsic apoptotic pathways in MDA-MB-231 and MDA-MB-468 TNBC cell lines when apoptosis-related genes were analyzed. 

TQ was found to upregulate 15 apoptotic genes in MDA-MB-231 TNBC cells, including caspases, GADD45A, TP53, DFFA, DIABLO, BNIP3, TRAF2/3, and TNFRSF10A. In MDA-MB-468 TNBC cells, 16 apoptotic genes were upregulated, including TNFRSF10A, TNF, TNFRSF11B, FADD TNFRSF10B, CASP2, and TRAF2, all of which are important for the apoptotic pathway, while suppressing the expression of one anti-apoptotic gene, BIRC5 in MDA-MB-231 cells. 

In order to preserve cellular homeostasis, apoptosis is regulated by caspases, or cysteinyl aspartate proteases [73]. A data analysis of the current study revealed that TQ influenced several caspases, including CASP2, CASP3, CASP4, and CASP9. TQ increased CASP3, 4, and 9 in MDA-MB-231 cells, with CASP9 being the most upregulated (+3.7). CASP2 is the only significantly upregulated caspase (+3.58) in MDA-MB-4688 cells (Table 3). TQ treatment stimulated CASP3 and CASP9 activities in MDA-MB-231 cells, which are regulators of both intrinsic and extrinsic (CASP3) or intrinsic (CASP9) apoptotic pathways. CASP2 and CASP9 are apoptosis initiator caspases that promote innate immunological responses to cellular stress and cause pyroptosis [74]. CASP3 is thought to cleave various structural and repair proteins to coordinate the execution phase of apoptosis [75]. CASP4 is an inflammatory caspase that activates cytokines, and it is suggested to play a role in pyroptosis, a type of cell death caused by pro-inflammatory signals that is linked to inflammation [76]. TQ’s anti-cancer effect on the MDA-MB-231 and MDA-MB-468 cells is thought to be mediated through pyroptosis processes, in addition to apoptosis. CASP2 is a unique member of the caspases family since it has both initiator and effector caspase characteristics. CASP2 is required for initiating the apoptotic process in response to various stressors, including DNA damage, TNF injection, and several infections and viruses [77]. According to our findings, TQ has multipurpose actions in regulating and generating initiator, effector, and inflammatory caspases, which effectively promote apoptosis in TNBC cells.

TQ significantly altered the expression of some genes belonging to the Bcl-2 family. Three genes that encode the pro-apoptotic proteins, BAD, BIK, and BAX [78], were found to be significantly upregulated in MDA-MB-468 cells. Meanwhile, BNIP3 and MCL1 were upregulated in MDA-MB-231 cells. Members of the Bcl-2 family regulate mitochondrial membrane permeability and are essential regulators of cell death and survival [79]. They measure the upregulated levels of BAD, BIK, and BAX in MDA-MB-468 cells, indicating that TQ causes a shift from anti-apoptosis to pro-apoptosis by altering the function of Bcl-2 family proteins. These three pro-apoptotic proteins initiate the intrinsic apoptosis pathway through the permeabilizing mitochondrial membrane, allowing apoptotic proteins such as CYCS to be released and the activation of caspases [80]. Further research has suggested that these proteins play a role in DNA damage-induced apoptosis and S-phase cell cycle arrest [81]. BNIP3, the most well-known member of BNIP, was altered in TQ-treated MDA-MB-231 cells. BNIP3 is required to activate the mitochondrial intrinsic apoptosis pathway [82] and has been shown to respond to hypoxia, a condition that activates Tp53 [83]. According to previous research, the overexpression of BNIP3 induces both caspase-dependent [82] and caspase-independent apoptosis, as well as autophagy [84] and necrosis-like cell death [85]. Furthermore, a link between increased BNIP3 expression, cell cycle arrest, and apoptosis was previously discovered [86]. MCL-1 gene expression is involved in polyubiquitination and proteasomal degradation, which promotes dynamic responses to cell death stimuli [87,88]. According to one study, MCL-1’s anti-proliferative function is clearly linked to its ability to bind proliferating cell nuclear antigen, which regulates the cell cycle; however, it is distinct from its anti-apoptotic activity [88]. Another study found a short form of MCL-1 in the nucleus that binds to and negatively regulates cdk1 activity [89]. In parallel, this finding could explain the cell cycle arrest in MDA-MB-231 TNBC cells in our study. Taken together, in the current studies, BAD, BIK, BAX, BNIP3, and MCL1 overexpression were found to contribute to TQ’s apoptotic effects in MDA-MB-231 and MDA-MB-468 TNBC cells and could be used as another potential target molecule by TQ in the treatment of TNBC patients.

Two caspase recruitment domain (CARD) family members, APAF1 and BCL10, were found to be significantly altered in TQ-treated MDA-MB-468 and MDA-MB-231 TNBC cells, respectively. BCL10 is a pro-apoptotic gene that mediates apoptosis by activating the apoptotic protease activating factor (APAF1)-caspase-9 pathway through a CASP8-independent mechanism [79,90]. Furthermore, as an immune signaling adaptor, BCL10 mediates immune responses and has the potential to cause cell cycle arrest [91]. Overexpression of this gene in BCL10-transfected BC cells was linked to apoptosis [92]. APAF1 (apoptotic protease activating factor) is essential for mitochondrial-mediated apoptosis. Cancer cells are widely protected from apoptosis due to low APAF1 levels compared to normal cells. Interestingly, our findings are consistent with other studies. Previous research found that MCF7 cells transfected with APAF1 were more sensitive to Docetaxel due to increasing the mitochondrial permeability for cytochrome c release, which promoted apoptosome formation and intrinsic apoptosis [93]. Taken together, inducing the expression of BCL10 and APF1 in TNBC cells could be partly responsible for TQ’s apoptotic effects.

TQ treatment caused mRNA expression upregulation in multiple TNFR family genes in both TNBC cell models. We also suggested that these two cell lines utilized a distinct mechanism to undergo apoptosis. TNF, TNFRSF10A, TNFRSF10B, TNFRSF11B, TNFRSF21, and FADD protein levels were significantly increased in TQ-treated MDA-MB-468 cells. TQ significantly increased the expression of TNF and tumor necrosis factor receptor superfamily (TNFRSF) members (TNFRSF10A, 10B,11B, and 21). The TNFRSF10A gene binds to TNF-α and causes apoptosis [94], whereas the TNFRSF10B gene is one of the receptors that can bind to a TNFSF10/TRAIL ligand [95] and cause apoptosis via caspase recruitment [96]. In TQ-treated MDA-MB-468 cells, TNFRSF10A is the most upregulated gene (27.67-fold). In contrast to healthy breast cells, low TNFRSF10A expression in various types of cancer, including BC, is associated with decreased apoptosis and cancer cell enhancement [97]. TNFRSF10A upregulation, on the other hand, mediates apoptosis in transformed cells, while having either no effect or a negligible effect on normal cells [98]. Previous research suggested that TNFRSF10A activates BID and participates in the intrinsic mitochondrial apoptotic pathway [99]. The upregulated TNFRSF10A can mediate cell cycle arrest, anti-proliferative and apoptotic effects, which is consistent with previous findings in prostate cancer cell line studies [100]. TNFRSF10A was also found to be significantly upregulated in MDA-MB-231 cells, albeit at a lower fold change than in MDA-MB-468 cells. As a result, our findings suggest that TNFSRF10A plays a significant role in the observed apoptotic effect in TQ-treated MDA-MB-468 and MDA-MB-231 cells and TNFRSF10A as a novel target in TNBC patients. On the other hand, the upregulated TNFRSF21 found in TQ-treated MDA-MB-468 cells was previously shown to induce apoptosis [101], possibly via the mitochondria-mediating pathway and Bax interaction [102]. As a result, our findings suggest that TNFRSF members play a significant role in the observed apoptotic effect in TQ-treated MDA-MB-468 and MDA-MB-231 cells and suggest a novel target in TNBC patients.

FADD receptor was also one of the most significantly upregulated genes in MDA-MB-468 cells (14.86-fold change, Table 3). This gene is involved in the extrinsic and TRAIL-induced apoptotic pathways via caspase activation [103] as well as playing a critical role in BID cleavage [75]. In MDA-MB-468, the upregulated expression of FADD is suggested to mediate the observed S-phase cell cycle arrest and apoptosis, consistent with previous BC findings [104]. 

TQ was found to upregulate the expression of TRAFs in MDA-MB-231 and MDA-MB-468 cells. TRAFs (tumor necrosis factor receptor-associated factors) are a group of structurally related proteins that transmit signals between members of TNFRSF and other immune receptors [105]. The transcription factor nuclear factor β (NF-κβ) and MAPKs are two major downstream signaling events regulated by TRAF molecules [106,107]. TRAF2 and TRAF3 are negative regulators of various signaling pathways, such as NF-kβ and pro-inflammatory Toll-like receptor (TLR) pathways [108]. TQ dramatically augmented the expression of TRAF2 and TRAF3 proteins by +2.8- and +2.3-fold, respectively. TQ’s overexpression of these two negative regulator proteins may play a role in the downregulation of NF-κβ. However, our hypothesis urges further investigation and could be a valuable therapeutic target for TNBC patients. TRAF2 mediates apoptosis in breast, cervical, and lung cancer cells by stabilizing the caspase-2 dimer complex and enhancing its activity to fully commit the cell to death, according to prior research [109,110]. In this study, TQ’s overexpression of these two negative regulator proteins may play a role in the apoptotic effects of TQ and could be a useful therapeutic target for TNBC patients.

The pro-apoptotic gene GADD45A, a Tp53 and DNA damage response gene, was the most upregulated in TQ-treated MDA-MB-231 cells. GADD45A (growth arrest and DNA damage-induced 45 alpha) is a pro-apoptotic gene belonging to the GADD45 stress sensor family. GADD45A is a critical component of several metabolic pathways that mediates cell proliferation, DNA repair, apoptosis, and cell cycle regulation [111]. The tumor suppressors Tp53 and BRCA1 use GADD45A as a transcriptional target. In various cancer cells, including BC, GADD45A repression was strongly associated with increased cancer cells survival, uncontrolled proliferation, and carcinogenesis [112,113]. GADD45A levels are modest in normal breast tissue, highest in Luminal A subtypes, followed by Luminal B subtypes and HER^+^ subtypes, and lowest in TNBC [114]. The absence of the three hormone receptors, ER, PR, and Her2/neu, is strongly linked to a low level of the GADD45A gene in TNBC [114]. With a significant GADD45a expression in BC progression and its ability to upregulate GADD45A and induce apoptosis, TQ is a promising therapeutic agent for managing TNBC cells, either directly or indirectly.

The tumor suppressor gene TP53, the most significantly altered gene in cancer [115], is one of the most upregulated genes in TQ-treated MDA-MB-231 cells. Many cell activities, including cell cycle regulation, senescence, survival, proliferation, and apoptosis, are regulated by the tumor suppressor protein encoded by the genome [116]. Certainly, Tp53 is essential for tumor suppression during the acute response to cellular stress, as well as the death or silencing of cancer-initiating cells with oncogenic lesions that drive neoplastic transformation [117]. Following cellular stress and DNA damage, Tp53 is immediately induced in healthy cells. It contributes to the transcriptional activation of several genes, including BCL2-associated X(BAX) [118] and GADD45A [119], which in turn stop cancer cell proliferation and delete or diminish cancer cell tumorigenic potential. Tp53 mutation is a turning point in the evolution of aggressive and metastatic BC with the worst prognosis. In TNBC, the frequency of Tp53 mutations is more than 50% [120], and inactivation of the Tp53 gene has a significant impact on TNBC prognosis [121]. According to our findings, TQ inhibits the proliferation of MDA-MB-231 and MDA-MB-468. Therefore, the upregulation of Tp53 expression (+2.5-fold change, Table 3) by TQ, which could be either a mutant or converted wild-type, was unknown in our study but may be suggested as a mediator in the profound antiproliferative effort, at least in part. Previous studies in Tp3-mutant TNBC cells demonstrated that TQ causes G1 phase cell cycle arrest and induces apoptosis [68]. Previous research also found that several compounds, including PRIMA-1, APR-246 PK11007, and COTI-2, can reactivate mutant p53 protein and convert it to a conformation with wild-type properties [122]. Nonenzymatically, APR-246 is converted to the Michael acceptor methylene quinuclidinone, which binds covalently to thiol groups in mutant p53, causing a reactivation of the mutant protein and induction of apoptosis [123]. Because TQ has quinone moiety in its structure, our findings of increased TP53 expression and induced apoptosis by TQ may be due to its conversion to wild-type properties, which require further investigation.

TQ significantly increased the expression of ACTB and BAG1 in both cell lines. Perhaps beta-actin helps to initiate apoptosis by allowing cytosolic pro-apoptotic proteins to be carried to mitochondria by a cytoskeleton-driven trafficking system [124]. Previous research has shown that actin can be used as a substrate for caspase cleavage in mammalian cells [125]. According to previous research, Bag1-L positively regulates the proapoptotic markers DP5 and Bim in HCT116 cells [126]. Other similar findings demonstrated that BAG1 enhanced radiation-induced apoptosis in squamous esophageal cancer cells (TE-12 cloneA1), which was further demonstrated by knocking down BAG1 expression using targeted siRNA that diminished radiosensitivity [127]. Taken together, TQ, by upregulating these two genes, may also contribute to the induction of apoptosis in MDA-MB-231 and MDA-MB-468 TNBC cells.

TQ-treated MDA-MB-231 cells exhibited a repressed expression of the BIRC (Baculoviral IAP Repeat Containing) family genes, which play a vital role in developing apoptosis resistance in various cancer cells, including BC cells [128]. Normally, BIRC5 was significantly altered. BIRC5 (survivin) is expressed in developing tissues, then disappears in adult cells except for a few cell types) before being re-expressed in malignancies [129]. BIRC5 is abundantly expressed in cancer tissue, particularly BC [130], and it has been shown to enhance cell proliferation in various malignancies. Additionally, BRIC5 has anti-apoptotic and mitotic regulating functions, allowing for anti-proteasomal stability and caspase-mediated apoptosis inhibition, increasing tumor growth and survival [129]. BIRC5 appears to be a unique prognostic factor and therapeutic target for TNBC. TQ significantly downregulates BIRC5 expression in MDA-MB-231TNBC cell lines, indicating a feasible alternative to the treatment and prevention of TNBC.

Despite some differences in gene expression caused by TQ in both cell lines, we believe that when present at pharmacological levels, TQ may affect the expression of genes involved in multiple apoptotic pathways in both cell lines.

## 5. Summary

The current study demonstrated the molecular mechanism underlying the apoptotic effect of the natural compound TQ in TNBC cells, MDA-MB-231, and MDA-MB-468 cells. TQ induced cytotoxic and anti-proliferative effects accompanied by apoptosis in response to S-phase and G2/M phase cell cycle arrest. Importantly, the variation in the molecular profiles between African American and Caucasian TNBC patients, including protein and gene expressions, somatic mutations, somatic DNA copy number alteration, and DNA methylation patterns [131], might explain the different genetic responses to TQ that were obtained in the two cell lines under investigation. Nonetheless, MDA-MB-468 cells were remarkably more susceptible to TQ than MDA-MB-231 cells. In MDA-MB-468 cells, many apoptosis-involved genes with a dramatically higher increase were measured. TQ was found to induce apoptosis in both TNBC cell models. As an in vitro investigation, our study has some limitations. While we explored the anti-cancer effects of TQ in two different BC cells, we did not explore normal breast cells or do any in vivo experiments in this study. Furthermore, the TNBC cell lines MDA-MB-468 and MDA-MB-231, also known as basal-like one and mesenchymal, are structurally distinct. Despite these drawbacks, this study offered explanations of the molecular pathways of TQ in two TNBC cell lines with different genetic backgrounds. The findings revealed that the potential of TQ to modify apoptosis-related factors significantly differed between the two cell lines. The results of this study should also propose TQ as an additional therapy for managing TNBC to improve chemotherapy effectiveness and limit cancer progression. In both MDA-MB-231 and MDA-MB-468 cells, TQ upregulates pro-apoptotic genes while suppressing the expression of the anti-apoptotic gene, BIRC5. Figure 10 summarizes the anti-cancer effect of TQ on genetically distinct TNBC cell lines.

## 6. Conclusions

The findings highlight the significance of the TQ as a candidate capable of inducing cancer cell apoptosis via intrinsic and extrinsic apoptosis-related genes. Our results related to the apoptotic gene profile revealed numerous helpful markers in these two TNBC cell lines, which may serve as a viable putative pharmaceutical target in TNBCs due to the aggressive character of the disease and the lack of focused therapies.

## Figures and Tables

**Figure 1 nutrients-14-02120-f001:**
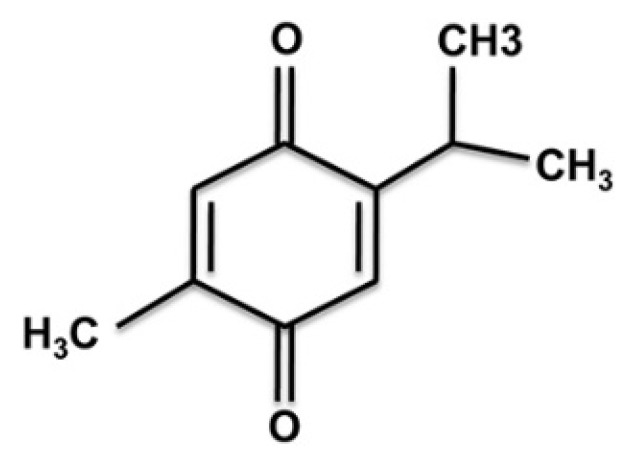
Thymoquinone’s chemical structure. Thymoquinone (2-isopropyl-5-methylbenzo-1, 4-quinone) is the main active ingredient in black seed volatile oil (*Nigella sativa* L.).

**Figure 2 nutrients-14-02120-f002:**
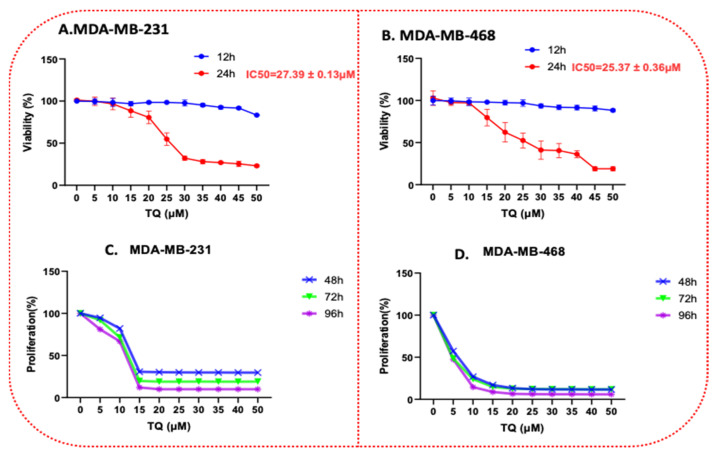
TQ reduces cell viability in TNBC cells using Alamar Blue. Both cell lines, MDA-MB231 and MDA-MB-468, were plated and treated similarly for 12 and 24 h in a time-dependent manner with TQ at concentration ranges of 0–50 μM (**A**,**B**). The graph shows the cell viability data expressed as percentages of cell survival compared to the control. Anti-proliferative effect of TQ in MDA-MB-231 I and MDA-MB-468 (**C**,**D**) TNBC cell lines. Both cell lines were seeded in 96-well plates at 1 × 10^4^ cells/100 µL/well and incubated for 48, 72, and 96 h with TQ at concentration ranges of 0 to 50 µM. Each data point presents the average ± SEM of three independent experiments, *n* = 6 each. One-way ANOVA was used to determine the significance of the difference between the control and treated groups.

**Figure 3 nutrients-14-02120-f003:**
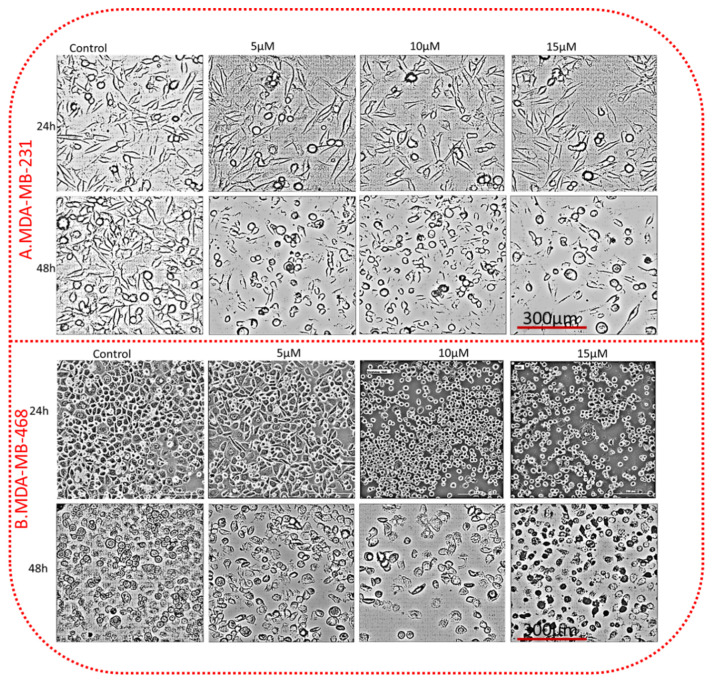
TQ alters the morphology of TNBC cells. Both cell lines were treated with 5, 10, and 15 µM TQ and control (only media with DMSO) for 24 h. The morphological changes in TQ-treated TNBC cells were visualized by a Cytation5 cell Imaging reader (BioTek Instruments, Inc., Winooski, VT, USA) using a phase-contrast digital microscope. Images represent MDA-MB-231 (**A**) and MDA-MB-468 (**B**). The treated cells showed irregular aggregates cells, small and round cells, apoptotic cells, and decreased cell density relative to control. All scale bars indicate 300 µm in 40× magnification.

**Figure 4 nutrients-14-02120-f004:**
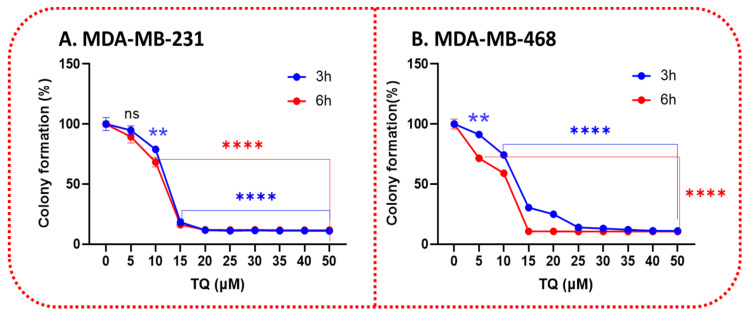
TQ treatment diminishes the clonogenic potential of TNBC cells. TQ’s colony formation effects over a prolonged period in MDA-MB-231 (**A**) and MDA-MB-468 (**B**) cells were shown by performing a colony formation assay. The TNBC cell lines were allowed to grow for seven days after treatment with 0–50 μM TQ for 3 and 6 h. Graphs were generated showing the relative colony formation ability of TNBC cells treated with TQ compared with controls. All error bars represent the standard error of the mean (*n* = 6). ** *p* < 0.01, **** *p* < 0.0001; ns, nonsignificant.

**Figure 5 nutrients-14-02120-f005:**
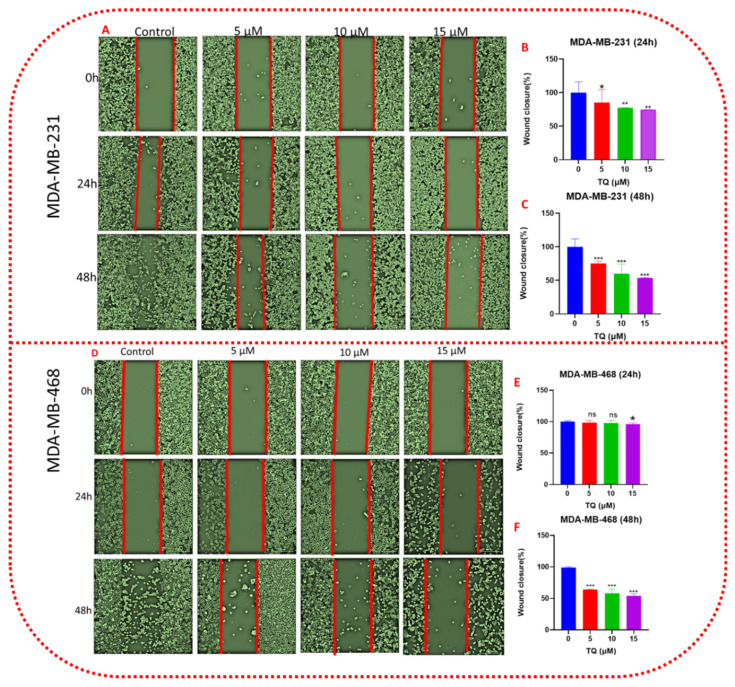
TQ attenuates TNBC cell migration. Figure 5 (**A**–**F**) shows a graphical representation of the wound-healing assay after 24 and 48 h in control and treated samples. Cell migration was assessed in MDA-MB-231 and MDA-MB-468 after exposure to TQ at concentrations of 0, 5, 10, and 15 µM. Both cells were treated with TQ for 24 h, and migration was assessed using a wound-healing assay in a 12-well plate (as described in Section 2). MDA-MB-231 (**A**) and MDA-MB-468 (**D**) typical phase-contrast pictures of the wound site of control and TQ-treated cultures at 0, 24, and 48 h after treatment. The bar graph presents the concentration–response curves of TQ-treated or control cultures at 24 h and 48 h after processing in ImageJ software, as well as the residual wound area of percent fold change in the wound closure area, (**B**,**C**) (MDA-MB-231), and (**E**,**F**) (MDA-MB-468). Data presented as the mean ± SEM of three independent experiments, each performed in triplicates, were analyzed using Student’s *t*-test, * *p* < 0.05, ** *p* < 0.01, *** *p* < 0.001, ns-nonsignificant). All scale bars indicate 300 µm in 10× magnification.

**Figure 6 nutrients-14-02120-f006:**
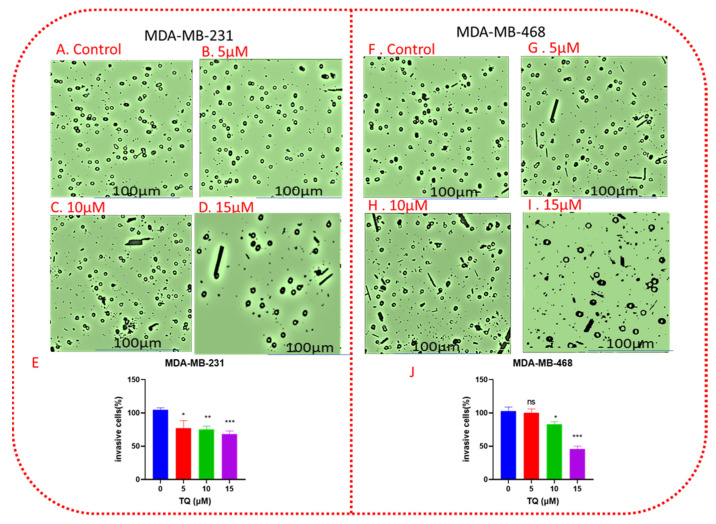
TQ inhibits TNBC cell invasion. A monolayer of TNBC MDA-MB-231 cells and MDA-MB-468 were treated with DMSO (control), 5, 10, and 15 µΜ for 24 h. Images (**A**–**D**,**F**–**I**) showed the relative invaded cells to the membrane for MDA-MB-231 and MDA-MB-468 cells, respectively, in a concentration dependent manner. Bar graphs (**E**,**J**) show the percentage of invaded cells per insert for the corresponding cell lines. The means ± SEMs of three independent experiments, each performed in triplicates, were analyzed by Student’s *t*-test. * *p* < 0.05, ** *p* < 0.01, *** *p* < 0.001; ns, nonsignificant. All scale bars indicate 100 µm in 10× magnification.

**Figure 7 nutrients-14-02120-f007:**
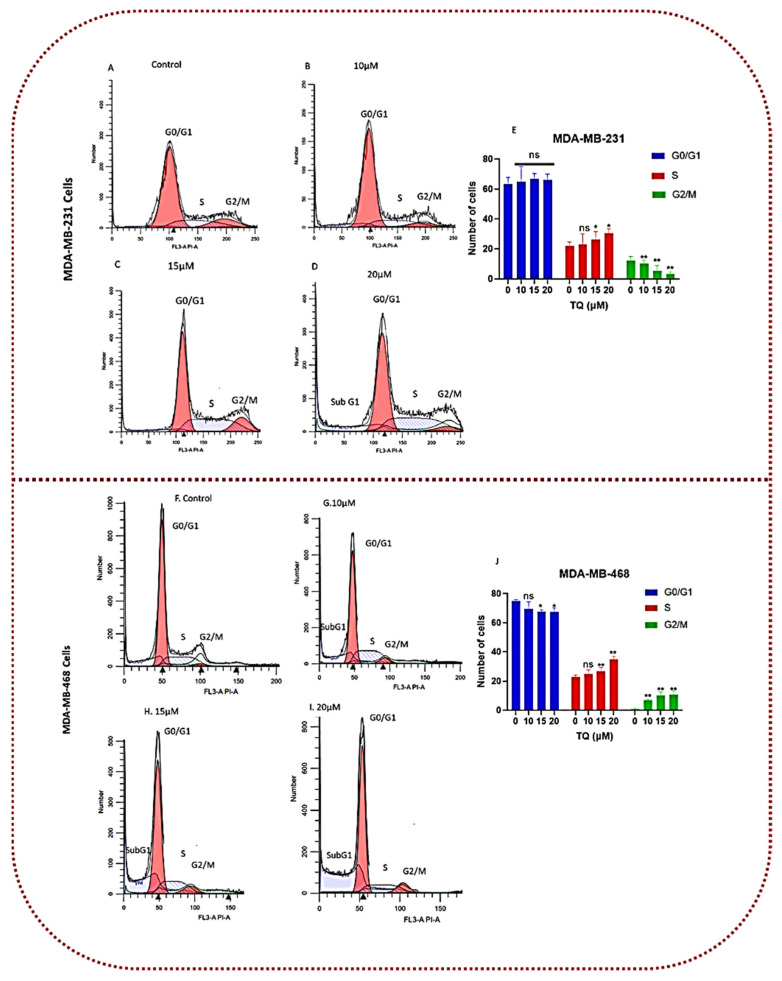
TQ affects cell cycle distribution in MDA-MB-231 and MDA-MB-468 TNBC cell lines. Both TNBC cell lines were treated for 24 h with TQ at three concentrations ranging from 0 to 20 µM. Sony SH800 cell sorter examination of cell dispersion using PI fluorescence. The histograms of representative Sony SH800 cell sorter experiments are shown (**A**–**D**) in MDA-MB-231 cells and (**F**–**I**) in MDA-MB-468 cells). Following three replicate studies, the bar graph shows (**E**,**J**) the average proportion of cells in each cell cycle phase. In MDA-MB-231 cells, TQ-treated TNBC cells showed an increase in the proportion of S phase cells and a decrease in the proportion of G2/M phase cells, while the proportion of S phase and G2/M increased in MDA-MB-468 cells. The *p*-value for the difference between the control and treated cells at different cell cycle phases was determined using one-way ANOVA followed by Bonferroni’s multiple comparisons test. At * *p* < 0.05, ** *p* < 0.01, the difference was declared significant and nonsignificant (ns).

**Figure 8 nutrients-14-02120-f008:**
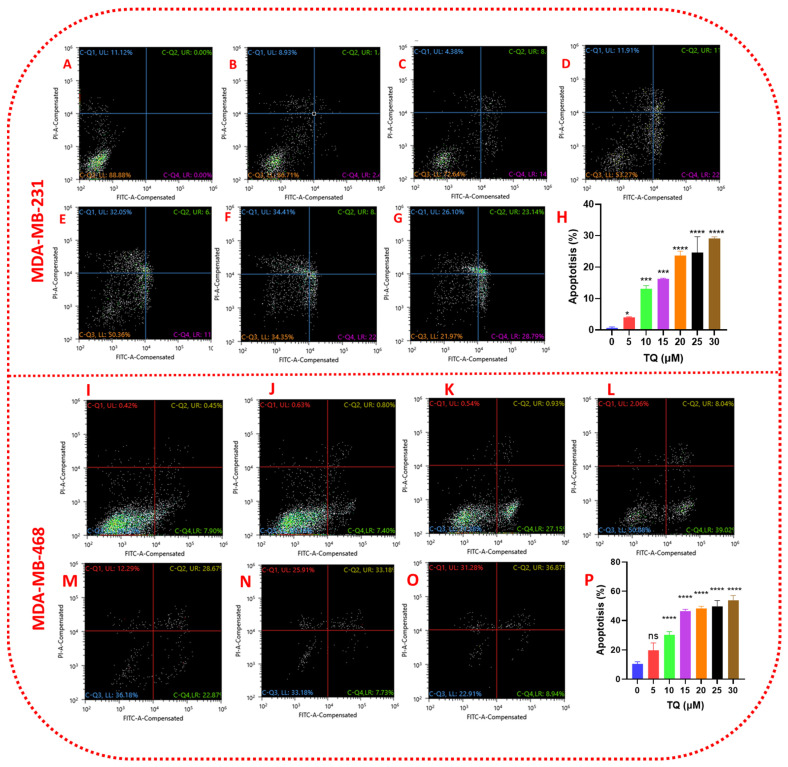
TQ induces apoptosis in TNBC cells. The apoptotic effect of TQ in TNBC cell lines is shown in (**A**–**H**) (MDA-MB-231) and (**I**–**P**) (MDA-MB-468). TQ was applied to both cell lines for 24 h at concentrations ranging from 0 to 30 µM. An experimental medium with DMSO was used to treat the control cells. A Sony SH800 cell sorter was used to evaluate the percentage of apoptotic cells in TQ-treated samples as well as the control, and an Annexin V-FITC apoptosis kit was utilized to mark treated/control cells. The mean ± S.E.M. of three independent investigations with *n* = 3 each is shown by each data point in the bar graphs. After TQ treatment, the percentage of apoptotic and necrotic cells increased, while the percentage of viable cells decreased. Using one-way ANOVA and Bonferroni’s multiple comparisons test, the significance of the difference between control and each treatment was assessed. The statistically significant differences were shown by, * *p* < 0.05, *** *p* < 0.001, and **** *p* < 0.0001, ns, nonsignificant.

**Figure 9 nutrients-14-02120-f009:**
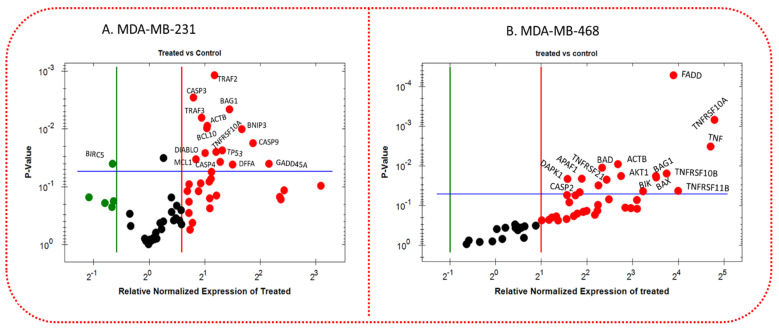
(**A**,**B**) TQ-altered apoptosis-related gene expression in MDA-MB-231 and MDA-MB-468 TNBC cells. After a 24 h treatment period with 27 µM µ in MDA-MB-231 cells (**A**) and 25 µM in MDA-MB-468 cells (**B**), a volcano plot was used to categorize and display increased(red), repressed (green), or unaltered (black) mRNA gene expression: actin, beta (ACTB), v-akt murine thymoma viral oncogene homolog 1 (AKT1), apoptotic peptidase activating factor 1 (APAF1), BCL2-associated agonist of cell death (BAD), BCL2-associated X protein (BAX), BCL2-associated athanogene 1 (BAG1), B-cell CLL/lymphoma 10 (BCL10), baculoviral IAP repeat containing 5 (BIRC5), BCL2-interacting killer (apoptosis-inducing) (BIK), BCL2/adenovirus E1B 19 kDa interacting protein 2 (BNIP3), death-associated protein kinase 1 (DAPK1), DNA fragmentation factor, 45 kDa, alpha polypeptide (DFFA), diablo, IAP-binding mitochondrial protein (DIABLO), growth arrest and DNA-damage-inducible, alpha (GADD45A), Caspase (CASP2,CASP3, CASP4, CASP9), Fas (TNFRSF6)-associated via death domain (FADD, myeloid cell leukemia sequence 1 (BCL2-related) (MCL1), tumor necrosis factor (TNF), tumor necrosis factor receptor superfamily, member 10A (TNFRSF10A), tumor necrosis factor receptor superfamily, member 10b (TNFRSF10B), tumor necrosis factor receptor superfamily, member 11b (TNFRSF11B), tumor necrosis factor receptor superfamily, member 21 (TNFRSF21), tumor protein p53 (TP53), TNF receptor-associated factor 2 (TRAF2), and TNF receptor-associated factor 3 (TRAF3). (**C**,**D**) RT-PCR analysis of TQ’s effect on specific apoptotic gene expression in MDA-MB-231 or MDA-MB-468 TNBC cell lines. TQ stimulates Caspases (CASP3,4, and 9), TRAF (2 and 3 and), BNIP3, BAG1, ACTB, BCL10, TP53, TNSFRSF10A, DIABLO, MCL1, DFFA, and GADD45A and inhibits BIRC5 gene expression in MDA-MB-231 TNBC cell lines after 27 µM of TQ treatment. TQ induction of FADD, TNF, TNSFRSF (10A,10B, 11B, and 21), ACTB, BAG1, TRAF2, BAX, DAPK1, APAF1, AKT1, BIK, and CASP2 genes in MDA-MB-468 TNBC cells after 25 µM TQ treatment. The control cells were treated with 0.1 percent DMSO. Gene expression was calculated using differences in mRNA expression between treatments and the control. The results are the mean SEM of at least four biological studies (*n* = 4). The significance of differences between the control and TQ treatments was determined using the student’s *t*-test The statistically significant differences were shown by * *p* < 0.05, ** *p* < 0.01, *** *p* < 0.001.

**Figure 10 nutrients-14-02120-f010:**
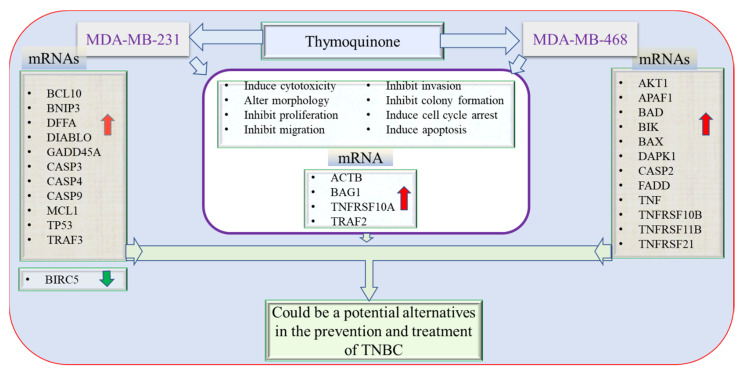
The effect of TQ on apoptotic gene expression in MDA-MB-231 and MDA-MB-468 TNBC cells. TQ inhibits proliferation, invasion, migration, and colony formation, as well as inducing cell cycle arrest and apoptosis. The red arrow indicates that apoptotic genes are being upregulated in both cells, while the green arrow indicates that one gene is being downregulated in MDA-MB-231 cells. Altogether, TQ may end up as a useful agent for the prevention and treatment of TNBC.

**Table 1 nutrients-14-02120-t001:** The mean distribution of cell cycle phases in TQ treated.

TQ (µM)	G0/G1 (% ± SEM)	S phase (% ± SEM)	G2/M (% ± SEM)
0	63.36 ± 4.5	21.95 ± 2.7	12.10 ± 2.8
10	65.05 ± 10.1	23.19 ± 6.8	10.26 ± 2.1 **
15	66.90 ± 3.5	26.36 ± 5.1 *	9.36 ± 4.7 **
20	66.01 ± 3.9	30.67 ± 2.7 *	3.33 ± 1.3 **

The *p*-value for the difference between the control and treated cells at different cell cycle phases was determined using one-way ANOVA followed by Bonferroni’s multiple comparisons test. At * *p* < 0.05 and ** *p* < 0.01.

**Table 2 nutrients-14-02120-t002:** The mean distribution of cell cycle phases in control TNBC cells.

TQ (µM)	G0/G1 (% ± SEM)	S phase (% ± SEM)	G2/M (% ± SEM)
0	75.04 ± 0.9	22.8 ± 1.4	0.9 ± 0.6
10	69.40 ± 5.1	25.1 ± 2.6	6.9 ± 0.6 **
15	67.60 ± 1.5 *	26.8 ± 2.4 **	10.3 ± 2.2 **
20	67.50 ± 2.7 *	34.7 ± 2.1 **	10.9 ± 2.4 **

The *p*-value for the difference between the control and treated cells at different cell cycle phases was determined using one-way ANOVA followed by Bonferroni’s multiple comparisons test. At * *p* < 0.05 and ** *p* < 0.01.

**Table 3 nutrients-14-02120-t003:** A comparative illustration of thymoquinone (TQ) effects on mRNA gene expression in MDA-MB-231 and MDA-MB-468 TNBC cells following a 24 h exposure period.

A. MDA-MB-231			B. MDA-MB-468		
Gene	Fold Change	*p*-Value	Gene	Fold Change	*p*-Value
*TRAF2*	+2.3	0.0011	*FADD*	+14.86	0.0005
*CASP3*	+1.8	0.0028	*TNFRSF10A*	+27.67	0.0006
*BAG1*	+2.7	0.0045	*TNF*	+26.07	0.0032
*TRAF3*	+1.9	0.0063	*ACTB*	+6.39	0.0089
*ACTB*	+2.1	0.0088	*BAD*	+5.03	0.0110
*BCL10*	+2.1	0.0088	*TNFRSF10B*	+11.39	0.0154
*BNIP3*	+3.2	0.0103	*BAG1*	+11.38	0.0178
*CASP9*	+3.7	0.0175	*TRAF2*	+11.42	0.0193
*TP53*	+2.5	0.0233	*BAX*	+6.71	0.0178
*TNFRSF10A*	+2.3	0.0248	*DAPK1*	+2.98	0.0285
*DIABLO*	+2.0	0.0259	*APAF1*	+3.69	0.0208
*MCL1*	+1.8	0.0332	*AKT1*	+5.39	0.0220
*CASP4*	+2.4	0.0233	*TNFRSF21*	+4.75	0.0308
*DFFA*	+2.8	0.0248	*TNFRSF11B*	+15.93	0.0422
*GADD45A*	+4.5	0.0398	*BIK*	+9.36	0.0432
*BIRC5*	−3.58	0.04398	*CASP2*	+3.58	0.0457

## Data Availability

All data generated or analyzed during this study are included in this published article.

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
