# Peer review of "Thymoquinone Alterations of the Apoptotic Gene Expressions and Cell Cycle Arrest in Genetically Distinct Triple-Negative Breast Cancer Cells"

_nutrients, 2022, doi:10.3390/nu14102120_

Round 1
Reviewer 1 Report
The article by Adinew and collegues gives an interesting insight about the activity of thymoquinone against tryple-negative breast cancer (TNBC). Several experiments, among which cytotoxicity studies, clonogenicity, cell cycle assay and apoptotic gene profiling, enrich the manuscript and deepen the effect of TQ in MDA-MB-468 and MDA-MB-231 cell lines.
The article is well written and organized.
- Authors should insert the structure of TQ in the introduction, not in the end of the manuscript.
- At page 6, line 280, correct “In contrast to MDA-MB-231 ..” with “In analogy with MDA-MB-231” since the cell viability is almost the same for both cell lines.
- At page 7, line 294 the concentrations are µM, not M. Furthermore, authors report that “subsequent experiments were carried out with 5, 10 and 15 µM to determine its biological effects” but different range of concentrations were actually used in further experiments. Hence, this phrase should be removed.
- Paragraph 3.3 should be merged with the 3.1, since they both analyze the proliferation rate of cell lines after TQ treatment.
- In table 3, letters A or B must be entered in the appropriate column.
- At page 19, lines 602-606 (concerning the viability results) should be inserted at the end of line 613, after describing TQ origin and introducing the experiments.
- At page 20, lines 640-642, some recent references about “compounds with the ability to alter oncogenic signaling pathway … eradicating cancer cells” are missing. Herein some examples: Eur J of Med Chem 237:114399 DOI: 10.1016/j.ejmech.2022.114399; ACS Med. Chem. Lett. 2022, 13, 3, 358-364 https://doi.org/10.1021/acsmedchemlett.1c00600; Eur J Med Chem 2022 Mar 16; 235:114292. doi: 10.1016/j.ejmech.2022.114292
- The appropriate citation is also missing at page 20, line 651.
- Please check that the marine species of origin is always written in italics and that “IC50” is always written correctly.
Author Response
Dear editor:
We are pleased to resubmit the revised version of the Manuscript ID: nutrients- 1715452. Title: " Thymoquinone Alterations of the Apoptotic Gene Expressions and Cell Cycle Arrest in Genetically Distinct Triple-Negative Breast Cancer Cells." We appreciate the reviewers' constructive criticisms, questions, and comments, and we have addressed each of their concerns as outlined below.
Reviewer 1
- Authors should insert the structure of TQ in the introduction, not at the end of the manuscript.
- Response: Corrected as suggested on page 2, lines 87-90.
- On page 6, line 280, correct "In contrast to MDA-MB-231." with "In analogy with MDA-MB-231" since the cell viability is the same for both cell lines.
- Response: Corrected as suggested (in analogy with MDA-MB-231) in line 284, page 6
- On page 7, line 294, the concentrations are µM, not M. Furthermore, the authors report that "subsequent experiments were carried out with 5, 10 and 15 µM to determine its biological effects," but a different range of concentrations was actually used in further experiments. Hence, this phrase should be removed.
- Response: Corrected as suggested. The phrase is removed on page 7, line 294
- Paragraph 3.3 should be merged with the 3.1 since they both analyze the proliferation rate of cell lines after TQ treatment.
- Response: Corrected as suggested included on pages 6-7, lines 293-317
- In table 3, letters A or B must be entered in the appropriate column.
- Response: Corrected as suggested included on page 16, line 537
- On page 19, lines 602-606 (concerning the viability results) should be inserted at the end of line 613 after describing TQ origin and introducing the experiments.
- Response: Corrected as suggested included on page 19 lines 505-508
- On page 20, lines 640-642, some recent references about "compounds with the ability to alter oncogenic signaling pathway … eradicating cancer cells" are missing. Herein some examples: Eur J of Med Chem 237:114399 DOI: 10.1016/j.ejmech.2022.114399; ACS Med. Chem. Lett. 2022, 13, 3, 358-364 https://doi.org/10.1021/acsmedchemlett.1c00600; Eur J Med Chem 2022 Mar 16; 235:114292. doi: 10.1016/j.ejmech.2022.114292
- Response: Corrected as suggested, included on page 20 lines 636-637
- The appropriate citation is also missing on page 20, line 651.
- Response: Corrected as suggested included on page 20 lines 643-645
- Please check that the marine species of origin is always written in italics and that "IC50" is always written correctly.
- Response: Checked and corrected as suggested and included in the manuscript

Reviewer 2 Report
The anticancer and proapoptotic properties of thymoquinone have been elucidated several years ago. In the current manuscript the authors demonstrated the general mechanism underlying the anticancer effect of the natural compound TQ in 2 different BC cells. The manuscript is interesting, well organized and clearly presented, but it's not very original.
I suggest the subsequent revision:
1- Section "2.1. Materials and Reagents": the authors could provide the catalog number of TQ.
2- Figure 2: it would be better to replace the figure 2 with a higher quality one.
4- Some references are more than 15 years old, is it possible to replace them with more current references?
Author Response
Reviewer two
Dear editor:
We are pleased to resubmit the revised version of the Manuscript ID: nutrients- 1715452. Title: " Thymoquinone Alterations of the Apoptotic Gene Expressions and Cell Cycle Arrest in Genetically Distinct Triple-Negative Breast Cancer Cells." We appreciate the reviewers' constructive criticisms, questions, and comments, and we have addressed each of their concerns as outlined below.
Comments and Suggestions for Authors:
The anticancer and proapoptotic properties of thymoquinone were elucidated several years ago. In the current manuscript, the authors demonstrated the general mechanism underlying the anticancer effect of the natural compound TQ in 2 different BC cells. The manuscript is interesting, well organized, and clearly presented, but it is not very original.
I suggest the subsequent revision:
- Section "2.1. Materials and Reagents": the authors could provide the catalog number of TQ.
- Response: Corrected as suggested, included on page 3, line 106
- Figure 2: it would be better to replace figure 2 with a higher-quality one.
- Response: Replaced as suggested, included on page 8, line 329
- Some references are more than 15 years old; is it possible to replace them with more current references?
- Response: Corrected as suggested, the reference cites are now starting from 2007 and included in each citation of the whole manuscript
